# DEEP BATCH ACTIVE LEARNING BY DIVERSE, UNCERTAIN GRADIENT LOWER BOUNDS

**Jordan T. Ash**
Princeton University

**Chicheng Zhang**
University of Arizona

**Akshay Krishnamurthy**
Microsoft Research NYC

**John Langford**
Microsoft Research NYC

**Alekh Agarwal**
Microsoft Research Redmond

## ABSTRACT

We design a new algorithm for batch active learning with deep neural network models. Our algorithm, Batch Active learning by Diverse Gradient Embeddings (BADGE), samples groups of points that are disparate and high magnitude when represented in a hallucinated gradient space, a strategy designed to incorporate both predictive uncertainty and sample diversity into every selected batch. Crucially, BADGE trades off between uncertainty and diversity without requiring any hand-tuned hyperparameters. While other approaches sometimes succeed for particular batch sizes or architectures, BADGE consistently performs as well or better, making it a useful option for real world active learning problems.

## 1 INTRODUCTION

In recent years, deep neural networks have produced state-of-the-art results on a variety of important supervised learning tasks. However, many of these successes have been limited to domains where large amounts of labeled data are available. A promising approach for minimizing labeling effort is *active learning*, a learning protocol where labels can be requested by the algorithm in a sequential, feedback-driven fashion. Active learning algorithms aim to identify and label only maximally-informative samples, so that a high-performing classifier can be trained with minimal labeling effort. As such, a robust active learning algorithm for deep neural networks may considerably expand the domains in which these models are applicable.

How should we design a practical, general-purpose, label-efficient active learning algorithm for deep neural networks? Theory for active learning suggests a version-space-based approach (Cohn et al., 1994; Balcan et al., 2006), which explicitly or implicitly maintains a set of plausible models, and queries examples for which these models make different predictions. But when using highly expressive models like neural networks, these algorithms degenerate to querying every example. Further, the computational overhead of training deep neural networks precludes approaches that update the model to best fit data after each label query, as is often done (exactly or approximately) for linear methods (Beygelzimer et al., 2010; Cesa-Bianchi et al., 2009). Unfortunately, the theory provides little guidance for these models.

One option is to use the network's uncertainty to inform a query strategy, for example by labeling samples for which the model is least confident. In a batch setting, however, this creates a pathological scenario where data in the batch are nearly identical, a clear inefficiency. Remedying this issue, we could select samples to maximize batch diversity, but this might choose points that provide little new information to the model.

For these reasons, methods that exploit just uncertainty or diversity do not consistently work well across model architectures, batch sizes, or datasets. An algorithm that performs well when using a ResNet, for

example, might perform poorly when using a multilayer perceptron. A diversity-based approach might work well when the batch size is very large, but poorly when the batch size is small. Further, what even constitutes a "large" or "small" batch size is largely a function of the statistical properties of the data in question. These weaknesses pose a major problem for real, practical batch active learning situations, where data are unfamiliar and potentially unstructured. There is no way to know which active learning algorithm is best to use.

Moreover, in a real active learning scenario, every change of hyperparameters typically causes the algorithm to label examples not chosen under other hyperparameters, provoking substantial labeling inefficiency. That is, hyperparameter sweeps in active learning can be label expensive. As a result, active learning algorithms need to "just work", given fixed hyperparameters, to a greater extent than is typical for supervised learning.

Based on these observations, we design an approach which creates diverse batches of examples about which the current model is uncertain. We measure uncertainty as the gradient magnitude with respect to parameters in the final (output) layer, which is computed using the most likely label according to the model. To capture diversity, we collect a batch of examples where these gradients span a diverse set of directions. More specifically, we build up the batch of query points based on these hallucinated gradients using the $k$-MEANS++ initialization (Arthur and Vassilvitskii, 2007), which simultaneously captures both the magnitude of a candidate gradient and its distance from previously included points in the batch. We name the resulting approach Batch Active learning by Diverse Gradient Embeddings (BADGE).

We show that BADGE is robust to architecture choice, batch size, and dataset, generally performing as well as or better than the best baseline across our experiments, which vary all of the aforementioned environmental conditions. We begin by introducing our notation and setting, followed by a description of the BADGE algorithm in Section 3 and experiments in Section 4. We defer our discussion of related work to Section 5.

## 2 NOTATION AND SETTING

Define $[K] := \{1, 2, \dots, K\}$. Denote by $\mathcal{X}$ the instance space and by $\mathcal{Y}$ the label space. In this work we consider multiclass classification, so $\mathcal{Y} = [K]$. Denote by $D$ the distribution from which examples are drawn, by $D_{\mathcal{X}}$ the unlabeled data distribution, and by $D_{\mathcal{Y}|\mathcal{X}}$ the conditional distribution over labels given examples. We consider the pool-based active learning setup, where the learner receives an unlabeled dataset $U$ sampled according to $D_{\mathcal{X}}$ and can request labels sampled according to $D_{\mathcal{Y}|\mathcal{X}}$ for any $x \in U$. We use $\mathbb{E}_D$ to denote expectation under the data distribution $D$. Given a classifier $h : \mathcal{X} \to \mathcal{Y}$, which maps examples to labels, and a labeled example $(x, y)$, we denote the $0/1$ error of $h$ on $(x, y)$ as $\ell_{01}(h(x), y) = I(h(x) \neq y)$. The performance of a classifier $h$ is measured by its expected $0/1$ error, i.e. $\mathbb{E}_D[\ell_{01}(h(x), y)] = \Pr_{(x,y) \sim D}(h(x) \neq y)$. The goal of pool-based active learning is to find a classifier with a small expected $0/1$ error using as few label queries as possible. Given a set $S$ of labeled examples $(x, y)$, where each $x \in S$ is picked from $U$, followed by a label query, we use $\mathbb{E}_S$ as the sample averages over $S$.

In this paper, we consider classifiers $h$ parameterized by underlying neural networks $f$ of fixed architecture, with the weights in the network denoted by $\theta$. We abbreviate the classifier with parameters $\theta$ as $h_\theta$ since the architectures are fixed in any given context, and our classifiers take the form $h_\theta(x) = \operatorname{argmax}_{y \in [K]} f(x; \theta)_y$, where $f(x; \theta) \in \mathbb{R}^K$ is a probability vector of scores assigned to candidate labels, given the example $x$ and parameters $\theta$. We optimize the parameters by minimizing the cross-entropy loss $\mathbb{E}_S[\ell_{\mathrm{CE}}(f(x; \theta), y)]$ over the labeled examples, where $\ell_{\mathrm{CE}}(p, y) = \sum_{i=1}^{K} I(y = i) \ln 1/p_i = \ln 1/p_y$.

---

**Algorithm 1** BADGE: Batch Active learning by Diverse Gradient Embeddings

---

**Require:** Neural network $f(x; \theta)$, unlabeled pool of examples $U$, initial number of examples $M$, number of iterations $T$, number of examples in a batch $B$.
 1: Labeled dataset $S \leftarrow M$ examples drawn uniformly at random from $U$ together with queried labels.
 2: Train an initial model $\theta_1$ on $S$ by minimizing $\mathbb{E}_S[\ell_{\mathrm{CE}}(f(x; \theta), y)]$.
 3: **for** $t = 1, 2, \ldots, T$: **do**
 4:     For all examples $x$ in $U \setminus S$:
            1.    Compute its hypothetical label $\hat{y}(x) = h_{\theta_t}(x)$.
            2.    Compute gradient embedding $g_x = \frac{\partial}{\partial \theta_{\mathrm{out}}} \ell_{\mathrm{CE}}(f(x; \theta), \hat{y}(x))|_{\theta = \theta_t}$, where $\theta_{\mathrm{out}}$ refers to parameters of the final (output) layer.
 5:     Compute $S_t$, a random subset of $U \setminus S$, using the $k$-MEANS++ seeding algorithm on $\{g_x : x \in U \setminus S\}$ and query for their labels.
 6:     $S \leftarrow S \cup S_t$.
 7:     Train a model $\theta_{t+1}$ on $S$ by minimizing $\mathbb{E}_S[\ell_{\mathrm{CE}}(f(x; \theta), y)]$.
 8: **end for**
 9: **return** Final model $\theta_{T+1}$.

---

## 3 ALGORITHM

BADGE, described in Algorithm 1, starts by drawing an initial set of $M$ examples uniformly at random from $U$ and asking for their labels. It then proceeds iteratively, performing two main computations at each step $t$: a *gradient embedding* computation and a *sampling* computation. Specifically, at each step $t$, for every $x$ in the pool $U$, we compute the label $\hat{y}(x)$ preferred by the current model, and the gradient $g_x$ of the loss on $(x, \hat{y}(x))$ with respect to the parameters of the last layer of the network. Given these gradient embedding vectors $\{g_x : x \in U\}$, BADGE selects a set of points by sampling via the $k$-MEANS++ initialization scheme (Arthur and Vassilvitskii, 2007). The algorithm queries the labels of these examples, retrains the model, and repeats.

We now describe the main computations — the embedding and sampling steps — in more detail.

**The gradient embedding.**    Since deep neural networks are optimized using gradient-based methods, we capture uncertainty about an example through the lens of gradients. In particular, we consider the model uncertain about an example if knowing the label induces a large gradient of the loss with respect to the model parameters and hence a large update to the model. A difficulty with this reasoning is that we need to know the label to compute the gradient. As a proxy, we compute the gradient as if the model's current prediction on the example is the true label. We show in Proposition 1 that, assuming a common structure satisfied by most natural neural networks, the gradient norm with respect to the last layer using this label provides a lower bound on the gradient norm induced by any other label. In addition, under that assumption, the length of this hypothetical gradient vector captures the uncertainty of the model on the example: if the model is highly certain about the example's label, then the example's gradient embedding will have a small norm, and vice versa for samples where the model is uncertain (see example below). Thus, the gradient embedding conveys information both about the model's uncertainty and potential update direction upon receiving a label at an example.

**The sampling step.**    We want the newly-acquired labeled samples to induce large and diverse changes to the model. To this end, we want the selection procedure to favor both sample magnitude and batch diversity. Specifically, we want to avoid the pathology of, for example, selecting a batch of $k$ similar samples where even just a single label could alleviate our uncertainty on all remaining $(k-1)$ samples.

A natural way of making this selection without introducing additional hyperparameters is to sample from a $k$-Determinantal Point Process ($k$-DPP; (Kulesza and Taskar, 2011)). That is, to select a batch of $k$ points

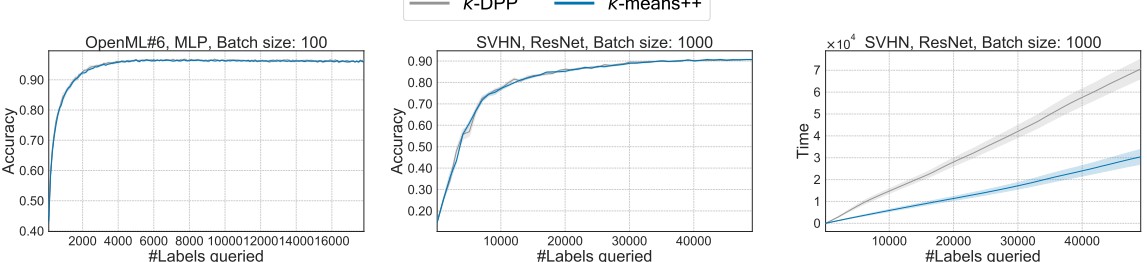

Figure 1: **Left and center**: Learning curves for $k$-MEANS++ and $k$-DPP sampling with gradient embeddings for different scenarios. The performance of the two sampling approaches nearly perfectly overlaps. **Right:** A run time comparison (seconds) corresponding to the middle scenario. Each line is the average over five independent experiments. Standard errors are shown by shaded regions.

with probability proportional to the determinant of their Gram matrix. Recently, Dereziński and Warmuth (2018) showed that in experimental design for least square linear regression settings, learning from samples drawn from a $k$-DPP can have much smaller mean square prediction error than learning from iid samples. In this process, when the batch size is very low, the selection will naturally favor points with a large length, which corresponds to uncertainty in our space. When the batch size is large, the sampler focuses more on diversity because linear independence, which is more difficult to achieve for large $k$, is required to make the Gram determinant non-zero.

Unfortunately, sampling from a $k$-DPP is not trivial. Many sampling algorithms (Kang, 2013; Anari et al., 2016) rely on MCMC, where mixing time poses a significant computational hurdle. The state-of-the-art algorithm of Dereziński (2018) has a high-order polynomial running time in the batch size and the embedding dimension. To overcome this computational hurdle, we suggest instead sampling using the $k$-MEANS++ seeding algorithm (Arthur and Vassilvitskii, 2007), originally made to produce a good initialization for $k$-means clustering. $k$-MEANS++ seeding selects centroids by iteratively sampling points in proportion to their squared distances from the nearest centroid that has already been chosen, which, like a $k$-DPP, tends to select a diverse batch of high-magnitude samples. For completeness, we give a formal description of the $k$-MEANS++ seeding algorithm in Appendix A.

**Example: multiclass classification with softmax activations.** Consider a neural network $f$ where the last nonlinearity is a softmax, i.e. $\sigma(z)_i = e^{z_i}/\sum_{j=1}^{K} e^{z_j}$. Specifically, $f$ is parametrized by $\theta = (W, V)$, where $\theta_{\text{out}} = W = (W_1, \ldots, W_K)^\top \in \mathbb{R}^{K \times d}$ are the weights of the last layer, and $V$ consists of weights of all previous layers. This means that $f(x; \theta) = \sigma(W \cdot z(x; V))$, where $z$ is the nonlinear function that maps an input $x$ to the output of the network's penultimate layer. Let us fix an unlabeled sample $x$ and define $p_i = f(x; \theta)_i$. With this notation, we have

$$\ell_{\text{CE}}(f(x; \theta), y) = \ln\left(\sum_{j=1}^{K} e^{W_j \cdot z(x;V)}\right) - W_y \cdot z(x; V).$$

Define $g_x^y = \frac{\partial}{\partial W}\ell_{\text{CE}}(f(x; \theta), y)$ for a label $y$ and $g_x = g_x^{\hat{y}}$ as the gradient embedding in our algorithm, where $\hat{y} = \text{argmax}_{i \in [K]} p_i$. Then the $i$-th block of $g_x$ (i.e. the gradients corresponding to label $i$) is

$$(g_x)_i = \frac{\partial}{\partial W_i}\ell_{\text{CE}}(f(x; \theta), \hat{y}) = (p_i - I(\hat{y} = i))z(x; V). \tag{1}$$

Based on this expression, we can make the following observations:

1. Each block of $g_x$ is a scaling of $z(x; V)$, which is the output of the penultimate layer of the network. In this respect, $g_x$ captures $x$'s representation information similar to that of Sener and Savarese (2018).

2. Proposition 1 below shows that the norm of $g_x$ is a lower bound on the norm of the loss gradient induced by the example with true label $y$ with respect to the weights in the last layer, that is $\|g_x\| \le \|g_x^y\|$. This suggests that the norm of $g_x$ conservatively estimates the example's influence on the current model.

3. If the current model $\theta$ is highly confident about $x$, i.e. vector $p$ is skewed towards a standard basis vector $e_j$, then $\hat{y} = j$, and vector $(p_i - I(\hat{y} = i))_{i=1}^{K}$ has a small length. Therefore, $g_x$ has a small length as well. Such high-confidence examples tend to have gradient embeddings of small magnitude, which are unlikely to be repeatedly selected by $k$-MEANS++ at iteration $t$.

**Proposition 1.** *For all $y \in \{1, \ldots, K\}$, let $g_x^y = \frac{\partial}{\partial W} \ell_{\text{CE}}(f(x; \theta), y)$. Then*

$$\|g_x^y\|^2 = \Big( \sum_{i=1}^{K} p_i^2 + 1 - 2p_y \Big) \|z(x; V)\|^2.$$

*Consequently, $\hat{y} = \operatorname{argmin}_{y \in [K]} \|g_x^y\|$.*

*Proof.* Observe that by Equation (1),

$$\|g_x^y\|^2 = \sum_{i=1}^{K} \big( p_i - I(y = i) \big)^2 \|z(x; V)\|^2 = \Big( \sum_{i=1}^{K} p_i^2 + 1 - 2p_y \Big) \|z(x; V)\|^2.$$

The second claim follows from the fact that $\hat{y} = \operatorname{argmax}_{y \in [K]} p_y$. □

This simple sampler tends to produce diverse batches similar to a $k$-DPP. As shown in Figure 1, switching between the two samplers does not affect the active learner's statistical performance but greatly improves its computational performance. Appendix G compares run time and test accuracy for both $k$-MEANS++ and $k$-DPP based sampling based on the gradient embeddings of the unlabeled examples.

Figure 2 illustrates the batch diversity and average gradient magnitude per selected batch for a variety of sampling strategies. As expected, both $k$-DPPs and $k$-MEANS++ tend to select samples that are diverse (as measured by the magnitude of their Gram determinant) and high magnitude. Other samplers, such as furthest-first traversal for $k$-Center clustering (FF-$k$-CENTER), do not seem to have this property. The FF-$k$-CENTER algorithm is the sampling choice of the CORESET approach to active learning, which we describe in the proceeding section (Sener and Savarese, 2018). Appendix F discusses diversity with respect to uncertainty-based approaches.

Appendix B provides further justification for why BADGE yields better updates than vanilla uncertainty sampling in the special case of binary logistic regression ($K = 2$ and $z(x; V) = x$).

## 4 EXPERIMENTS

We evaluate the performance of BADGE against several algorithms from the literature. In our experiments, we seek to answer the following question: How robust are the learning algorithms to choices of neural network architecture, batch size, and dataset?

To ensure a comprehensive comparison among all algorithms, we evaluate them in a batch-mode active learning setup with $M = 100$ being the number of initial random labeled examples and batch size $B$ varying from $\{100, 1000, 10000\}$. The following is a list of the baseline algorithms evaluated; the first performs representative sampling, the next three are uncertainty based, the fifth is a hybrid of representative and uncertainty-based approaches, and the last is traditional supervised learning.

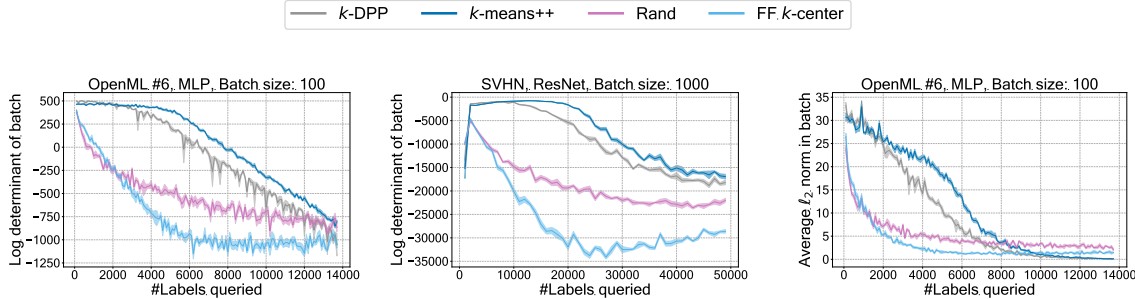

Figure 2: A comparison of batch selection algorithms using our gradient embedding. **Left and center:** Plots showing the log determinant of the Gram matrix of the selected batch of gradient embeddings as learning progresses. **Right:** The average embedding magnitude (a measurement of predictive uncertainty) in the selected batch. The FF-$k$-CENTER sampler finds points that are not as diverse or high-magnitude as other samplers. Notice also that $k$-MEANS++ tends to actually select samples that are both more diverse and higher-magnitude than a $k$-DPP, a potential pathology of the $k$-DPP's degree of stochastisity. Standard errors are shown by shaded regions.

1. CORESET: A diversity-based approach using coreset selection. The embedding of each example is computed by the network's penultimate layer and the samples at each round are selected using a greedy furthest-first traversal conditioned on all labeled examples (Sener and Savarese, 2018).

2. CONF (Confidence Sampling): An uncertainty-based active learning algorithm that selects $B$ examples with smallest predicted class probability, $\max_{i=1}^{K} f(x; \theta)_i$ (e.g. Wang and Shang, 2014).

3. MARG (Margin Sampling): An uncertainty-based active learning algorithm that selects the bottom $B$ examples sorted according to the example's multiclass margin, defined as $f(x; \theta)_{\hat{y}} - f(x; \theta)_{y'}$, where $\hat{y}$ and $y'$ are the indices of the largest and second largest entries of $f(x; \theta)$ (Roth and Small, 2006).

4. ENTROPY: An uncertainty-based active learning algorithm that selects the top $B$ examples according to the entropy of the example's predictive class probability distribution, defined as $H((f(x; \theta)_y)_{y=1}^{K})$, where $H(p) = \sum_{i=1}^{K} p_i \ln 1/p_i$ (Wang and Shang, 2014).

5. ALBL (Active Learning by Learning): A bandit-style meta-active learning algorithm that selects between CORESET and CONF at every round (Hsu and Lin, 2015).

6. RAND: The naive baseline of randomly selecting $k$ examples to query at each round.

We consider three neural network architectures: a two-layer Perceptron with ReLU activations (MLP), an 18-layer convolutional ResNet (He et al., 2016), and an 11-layer VGG network (Simonyan and Zisserman, 2014). We evaluate our algorithms using three image datasets, SVHN (Netzer et al., 2011), CIFAR10 (Krizhevsky, 2009) and MNIST (LeCun et al., 1998) [1], and four non-image datasets from the OpenML repository (#6, #155, #156, and #184). [2] We study each situation with 7 active learning algorithms, including BADGE, making for 231 total experiments.

For the image datasets, the embedding dimensionality in the MLP is 256. For the OpenML datasets, the embedding dimensionality of the MLP is 1024, as more capacity helps the model fit training data. We fit

---

[1]Because MNIST is a dataset that is extremely easy to classify, we only use MLPs, rather than convolutional networks, to better study the differences between active learning algorithms.

[2]The OpenML datasets are from openml.org and are selected on two criteria: first, they have at least 10000 samples; second, neural networks have a significantly smaller test error rate when compared to linear models.

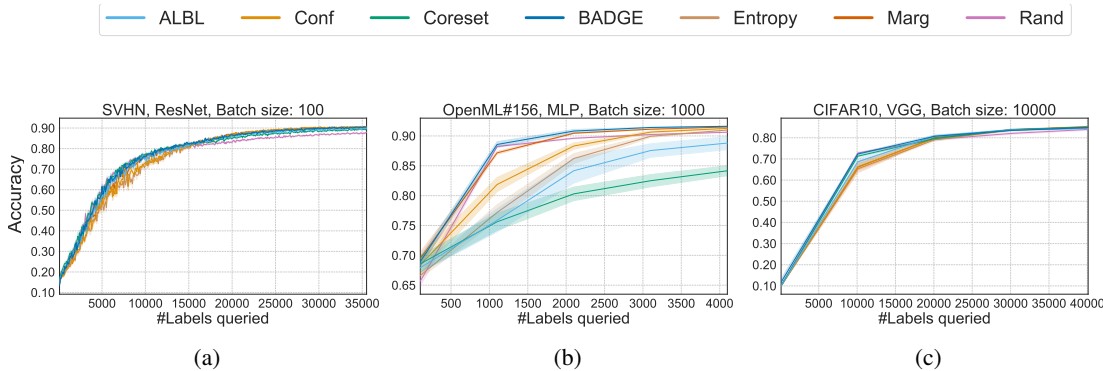

Figure 3: Active learning test accuracy versus the number of total labeled samples for a range of conditions. Standard errors are shown by shaded regions.

models using cross-entropy loss and the Adam variant of SGD until training accuracy exceeds 99%. We use a learning rate of $0.001$ for image data and of $0.0001$ for non-image data. We avoid warm starting and retrain models from scratch every time new samples are queried (Ash and Adams, 2019). All experiments are repeated five times. No learning rate schedules or data augmentation are used. Baselines use implementations from the libact library (Yang et al., 2017). All models are trained in PyTorch (Paszke et al., 2017).

**Learning curves.** Here we show examples of learning curves that highlight some of the phenomena we observe related to the fragility of active learning algorithms with respect to batch size, architecture, and dataset.

Often, we see that in early rounds of training, it is better to do diversity sampling, and later in training, it is better to do uncertainty sampling. This kind of event is demonstrated in Figure 3a, which shows CORESET outperforming confidence-based methods at first, but then doing worse than these methods later on.

In this figure, BADGE performs as well as diversity sampling when that strategy does best, and as well as uncertainty sampling once those methods start outpacing CORESET. This suggests that BADGE is a good choice regardless of labeling budget.

Separately, we notice that diversity sampling only seems to work well when either the model has good architectural priors (inductive biases) built in, or when the data are easy to learn. Otherwise, penultimate layer representations are not meaningful, and diverse sampling can be deleterious. For this reason, CORESET often performs worse than random on sufficiently complex data when not using a convolutional network (Figure 3b). That is, the diversity induced by unconditional random sampling can often yield a batch that better represents the data. Even when batch size is large and the model has helpful inductive biases, the uncertainty information in BADGE can give it an advantage over pure diversity approaches (Figure 3c). Comprehensive plots of this kind, spanning architecture, dataset, and batch size are in Appendix C.

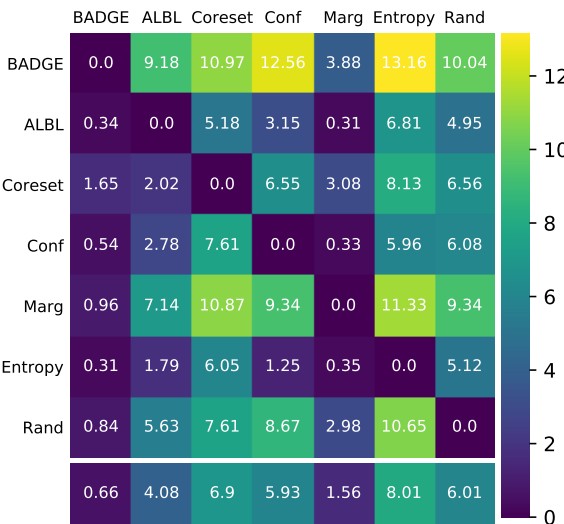

Figure 4: A pairwise penalty matrix over all experiments. Element $P_{i,j}$ corresponds roughly to the number of times algorithm $i$ outperforms algorithm $j$. Column-wise averages at the bottom show overall performance (lower is better).

**Pairwise comparisons.** We next show a comprehensive pairwise comparison of algorithms over all datasets ($D$), batch sizes ($B$), model architectures ($A$), and label budgets ($L$). From the learning curves, it can be observed that when label budgets are large enough, all algorithms eventually reach similar performance, making the comparison between them uninteresting in the large sample limit. For this reason, for each combination of $(D, B, A)$, we select a set of labeling budgets $L$ where learning is still progressing. We experimented with three different batch sizes and eleven dataset-architecture pairs, making the total number of $(D, B, A)$ combinations $3 \times 11 = 33$. Specifically, we compute $n_0$, the smallest number of labels where RAND's accuracy reaches 99% of its final accuracy, and choose label budget $L$ from $\left\{ M + 2^{m-1}B : m \in [\lfloor \log((n_0 - M)/B) \rfloor] \right\}$. The calculation of scores in the penalty matrix $P$ follows the following protocol: For each $(D, B, A, L)$ combination and each pair of algorithms $(i, j)$, we have 5 test errors (one for each repeated run), $\left\{e_i^1, \ldots, e_i^5\right\}$ and $\left\{e_j^1, \ldots, e_j^5\right\}$ respectively. We compute the $t$-score as $t = \sqrt{5}\hat{\mu}/\hat{\sigma}$, where

$$\hat{\mu} = \frac{1}{5}\sum_{l=1}^{5}(e_i^l - e_j^l), \qquad \hat{\sigma} = \sqrt{\frac{1}{4}\sum_{l=1}^{5}(e_i^l - e_j^l - \hat{\mu})^2}.$$

We use the two-sided $t$-test to compare pairs of algorithms: algorithm $i$ is said to *beat* algorithm $j$ in this setting if $t > 2.776$ (the critical point of $p$-value being 0.05), and similarly algorithm $j$ beats algorithm $i$ if $t < -2.776$. For each $(D, B, A)$ combination, suppose there are $n_{D,B,A}$ different values of $L$. Then, for each $L$, if algorithm $i$ beats algorithm $j$, we accumulate a penalty of $1/n_{D,B,A}$ to $P_{i,j}$; otherwise, if algorithm $j$ beats algorithm $i$, we accumulate a penalty of $1/n_{D,B,A}$ to $P_{j,i}$. The choice of the penalty value $1/n_{D,B,A}$ is to ensure that every $(D, B, A)$ combination is assigned equal influence in the aggregated matrix. Therefore, the largest entry of $P$ is at most 33, the total number of $(D, B, A)$ combinations.

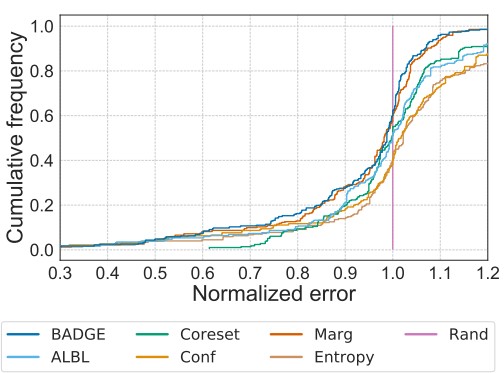

Figure 5: The cumulative distribution function of normalized errors for all acquisition functions.

Intuitively, each row $i$ indicates the number of settings in which algorithm $i$ beats other algorithms and each column $j$ indicates the number of settings in which algorithm $j$ is beaten by another algorithm.

The penalty matrix in Figure 4 summarizes all experiments, showing that BADGE generally outperforms baselines. Matrices grouped by batch size and architecture in Appendix D show a similar trend.

**Cumulative distribution functions of normalized errors.** For each $(D, B, A, L)$ combination, we compute the average error for each algorithm $i$ as $\bar{e}_i = \frac{1}{5}\sum_{l=1}^{5}e_i^l$. To ensure that the errors of these algorithms are on the same scale in all settings, we compute the normalized error of every algorithm $i$, defined as $\text{ne}_i = \bar{e}_i/\bar{e}_r$, where $r$ is the index of the RAND algorithm. By definition, the normalized errors of the RAND algorithm are identically 1 in all settings. Like with penalty matrices, for each $(D, B, A)$ combination, we only consider a subset of $L$ values from the set $\left\{ M + 2^{m-1}B : m \in [\lfloor \log((n_0 - M)/B) \rfloor] \right\}$. We assign a weight proportional to $1/n_{D,B,A}$ to each $(D, B, A, L)$ combination, where there are $n_{D,B,A}$ different $L$ values for this combination of $(D, B, A)$. We then plot the cumulative distribution functions (CDFs) of the normalized errors of all algorithms: for a value of $x$, the $y$ value is the total weight of settings where the algorithm has normalized error at most $x$; in general, an algorithm that has a higher CDF value has better performance.

We plot the generated CDFs in Figures 5, 22 and 23. We can see from Figure 5 that BADGE has the best overall performance. In addition, from Figures 22 and 23 in Appendix E, we can conclude that when batch size is small (100 or 1000) or when an MLP is used, both BADGE and MARG perform best. However, in the regime when the batch size is large (10000), MARG's performance degrades, while BADGE, ALBL and CORESET are the best performing approaches.

## 5 RELATED WORK

Active learning is a been well-studied problem (Settles, 2010; Dasgupta, 2011; Hanneke, 2014). There are two major strategies for active learning—representative sampling and uncertainty sampling.

Representative sampling algorithms select batches of unlabeled examples that are representative of the unlabeled set to ask for labels. It is based on the intuition that the sets of representative examples chosen, once labeled, can act as a surrogate for the full dataset. Consequently, performing loss minimization on the surrogate suffices to ensure a low error with respect to the full dataset. In the context of deep learning, Sener and Savarese (2018); Geifman and El-Yaniv (2017) select representative examples based on core-set construction, a fundamental problem in computational geometry. Inspired by generative adversarial learning, Gissin and Shalev-Shwartz (2019) select samples that are maximally indistinguishable from the pool of unlabeled examples.

On the other hand, uncertainty sampling is based on a different principle—to select new samples that maximally reduce the uncertainty the algorithm has on the target classifier. In the context of linear classification, Tong and Koller (2001); Schohn and Cohn (2000); Tur et al. (2005) propose uncertainty sampling methods that query examples that lie closest to the current decision boundary. Some uncertainty sampling approaches have theoretical guarantees on statistical consistency (Hanneke, 2014; Balcan et al., 2006). Such methods have also been recently generalized to deep learning. For instance, Gal et al. (2017) use Dropout as an approximation of the posterior of the model parameters, and develop information-based uncertainty reduction criteria; inspired by recent advances on adversarial examples generation, Ducoffe and Precioso (2018) use the distance between an example and one of its adversarial examples as an approximation of its distance to the current decision boundary, and uses it as the criterion of label queries. An ensemble of classifiers could also be used to effectively estimate uncertainty (Beluch et al., 2018).

There are several existing approaches that support a hybrid of representative sampling and uncertainty sampling. For example, Baram et al. (2004); Hsu and Lin (2015) present meta-active learning algorithms that can combine the advantages of different active learning algorithms. Inspired by expected loss minimization, Huang et al. (2010) develop label query criteria that balances between the representativeness and informativeness of examples. Another method for this is Active Learning by Learning (Hsu and Lin, 2015), which can select whether to exercise a diversity based algorithm or an uncertainty based algorithm at each round of training as a sequential decision process.

There is also a large body of literature on batch mode active learning, where the learner is asked to select a batch of samples within each round (Guo and Schuurmans, 2008; Wang and Ye, 2015; Chen and Krause; Wei et al., 2015; Kirsch et al., 2019). In these works, batch selection is often formulated as an optimization problem with objectives based on (upper bounds of) average log-likelihood, average squared loss, etc.

A different query criterion based on expected gradient length (EGL) has been proposed in the as well (Settles et al., 2008). In recent work, Huang et al. (2016) show that the EGL criterion is related to the $T$-optimality criterion in experimental design. They further demonstrate that the samples selected by EGL are very different from those by entropy-based uncertainty criterion. Zhang et al. (2017a) use the EGL criterion in active sentence and document classification with CNNs. These approaches differ most substantially from BADGE in that they do not take into account the diversity of the examples queried within each batch.

There is a wide array of theoretical articles that focus on the related problem of adaptive subsampling for fully-labeled datasets in regression settings (Han et al., 2016; Wang et al., 2018; Ting and Brochu, 2018). Empirical studies of batch stochastic gradient descent also employ adaptive sampling to "emphasize" hard or representative examples (Zhang et al., 2017b; Chang et al., 2017). These works aim at reducing computation costs or finding a better local optimal solution, as opposed to reducing label costs. Nevertheless, our work is inspired by their sampling criteria, which also emphasize samples that induce large updates to the model.

As mentioned earlier, our sampling criterion has resemblance to sampling from $k$-determinantal point processes (Kulesza and Taskar, 2011). Note that in multiclass classification settings, our gradient-based embedding of an example can be viewed as the outer product of the original embedding in the penultimate layer and a probability score vector that encodes the uncertainty information on this example (see Section 3). In this view, the penultimate layer embedding characterizes the diversity of each example, whereas the probability score vector characterizes the quality of each example. The $k$-DPP is also a natural probabilistic tool for sampling that trades off between quality and diversity (See Kulesza et al., 2012, Section 3.1). We remark that concurrent to our work, Bıyık et al. (2019) develops $k$-DPP based active learning algorithms based on this principle by explicitly designing diversity and uncertainty measures.

## 6 DISCUSSION

We have established that BADGE is empirically an effective deep active learning algorithm across different architectures and batch sizes, performing similar to or better than other active learning algorithms. A fundamental remaining question is: "Why?" While deep learning is notoriously difficult to analyze theoretically, there are several intuitively appealing properties of BADGE:

1. The definition of uncertainty (a lower bound on the gradient magnitude of the last layer) guarantees some update of parameters.

2. It optimizes for diversity as well as uncertainty, eliminating a failure mode of choosing many identical uncertain examples in a batch, and does so without requiring any hyperparameters.

3. The randomization associated with the $k$-MEANS++ initialization sampler implies that, even for adversarially constructed datasets, it eventually converges to a good solution.

The combination of these properties appears to generate the robustness that we observe empirically.

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

## A  THE $k$-MEANS++ SEEDING ALGORITHM

Here we briefly review the $k$-MEANS++ seeding algorithm by (Arthur and Vassilvitskii, 2007). Its basic idea is to perform sequential sampling of $k$ centers, where each new center is sampled from the ground set with probability proportional to the squared distance to its nearest center. It is shown in (Arthur and Vassilvitskii, 2007) that the set of centers returned is guaranteed to approximate the $k$-means objective function in expectation, thus ensuring diversity.

---

**Algorithm 2** The $k$-MEANS++ seeding algorithm (Arthur and Vassilvitskii, 2007)

---

**Require:** Ground set $G \subset \mathbb{R}^d$, target size $k$.
**Ensure:** Center set $C$ of size $k$.
  $C_1 \leftarrow \{c_1\}$, where $c_1$ is sampled uniformly at random from $G$.
  **for** $t = 2, \ldots, k$: **do**
    Define $D_t(x) := \min_{c \in C_{t-1}} \|x - c\|_2$.
    $c_t \leftarrow$ Sample $x$ from $G$ with probability $\frac{D_t(x)^2}{\sum_{x \in G} D_t(x)^2}$.
    $C_t \leftarrow C_{t-1} \cup \{c_t\}$.
  **end for**
  **return** $C_k$.

---

## B  BADGE FOR BINARY LOGISTIC REGRESSION

We consider instantiating BADGE for binary logistic regression, where $\mathcal{Y} = \{-1, +1\}$. Given a linear classifier $w$, we define the predictive probability of $w$ on $x$ as $p_w(y|x, \theta) = \sigma(yw \cdot x)$, where $\sigma(z) = \frac{1}{1+e^{-z}}$ is the sigmoid funciton.

Recall that $\hat{y} = \hat{y}(x)$ is the hallucinated label:

$$\hat{y}(x) = \begin{cases} +1, & p_w(+1|x, \theta) > 1/2, \\ -1, & p_w(+1|x, \theta) \le 1/2. \end{cases}$$

The binary logistic loss of classifier $w$ on example $(x, y)$ is defined as:

$$\ell(w, (x, y)) = \ln(1 + \exp(-yw \cdot x)).$$

Now, given model $w$ and example $x$, we define $\hat{g}_x = \frac{\partial}{\partial w} \ell(w, (x, \hat{y})) = (1 - p_w(\hat{y}|x, \theta)) \cdot (-\hat{y} \cdot x)$ as the loss gradient induced by the example with hallucinated label, and $\tilde{g}_x = \frac{\partial}{\partial w} \ell(w, (x, y)) = (1 - p_w(y|x, \theta)) \cdot (-y \cdot x)$ as the loss gradient induced by the example with true label.

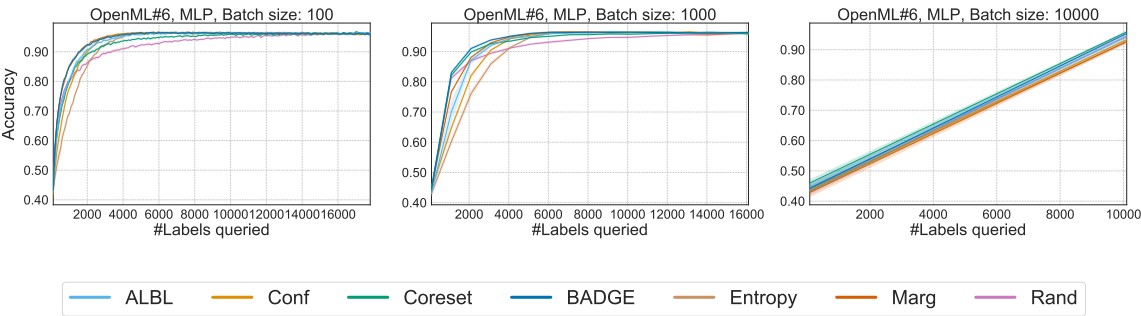

Figure 6: Full learning curves for OpenML #6 with MLP.

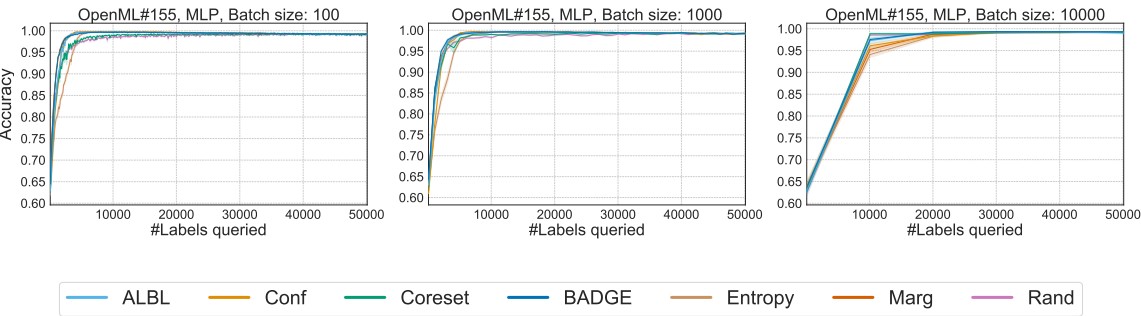

Figure 7: Full learning curves for OpenML #155 with MLP.

Suppose that BADGE only selects examples from region $S_w = \{x : w \cdot x = 0\}$, then as $p_w(+1|x,\theta) = p_w(-1|x,\theta) = \frac{1}{2}$, we have that for all $x$ in $S_w$, $\hat{g}_x = s_x \cdot g_x$ for some $s_x \in \{\pm 1\}$. This implies that, sampling from a DPP induced by $\hat{g}_x$'s is equivalent to sampling from a DPP induced by $g_x$'s. It is noted in Mussmann and Liang (2018) that uncertainty sampling (i.e. sampling from $D_{|S_w}$) implicitly performs preconditioned stochastic gradient descent on the expected 0-1 loss. In addition, it has been shown that DPP sampling over gradients may reduce the variance of the mini-batch stochastic gradient updates (Zhang et al., 2017b); this suggests that BADGE, when restricted its sampling over low-margin regions ($S_w$), improves over uncertainty sampling by collecting examples that together induce lower-variance updates on the gradient direction of expected 0-1 loss.

## C  ALL LEARNING CURVES

We plot all learning curves (test accuracy as a function of the number of labeled example queried) in Figures 6 to 12. In addition, we zoom into regions of the learning curves that discriminates the performance of all algorithms in Figures 13 to 19.

## D  PAIRWISE COMPARISONS OF ALGORITHMS

In addition to Figure 4 in the main text, we also provide penalty matrices (Figures 20 and 21), where the results are aggregated by conditioning on a fixed batch size (100, 1000 and 10000) or on a fixed neural network model (MLP, ResNet and VGG). For each penalty matrix, the parenthesized number in its title is the

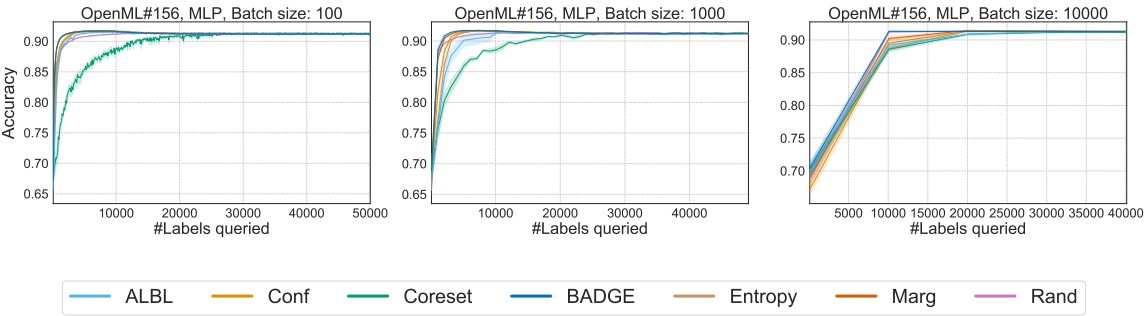

Figure 8: Full learning curves for OpenML #156 with MLP.

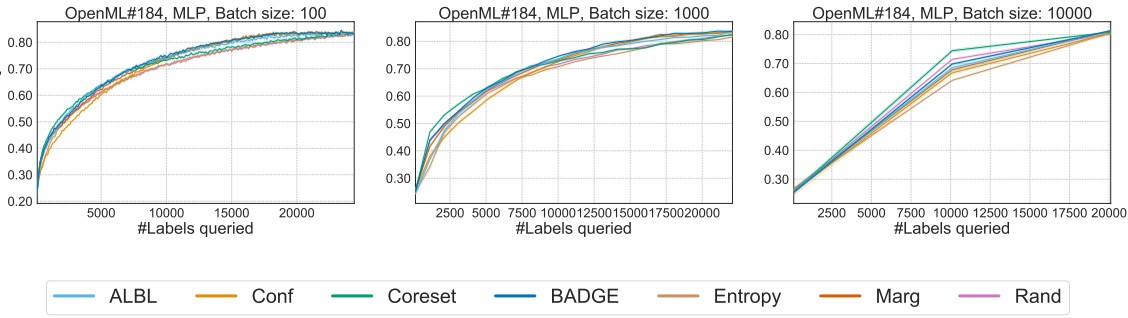

Figure 9: Full learning curves for OpenML #184 with MLP.

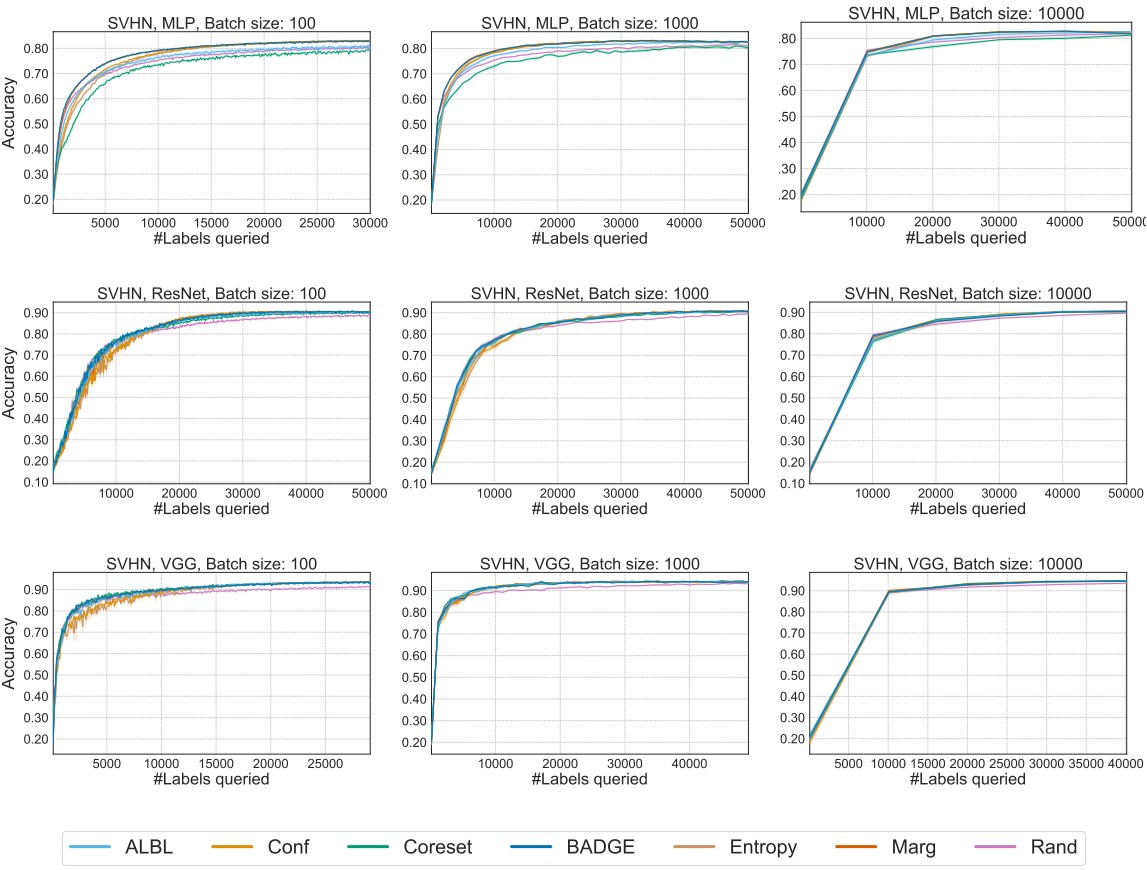

Figure 10: Full learning curves for SVHN with MLP, ResNet and VGG.

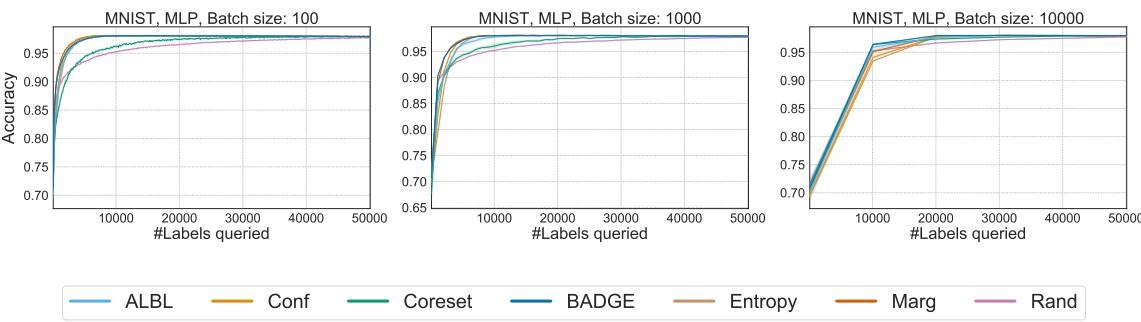

Figure 11: Full learning curves for MNIST with MLP.

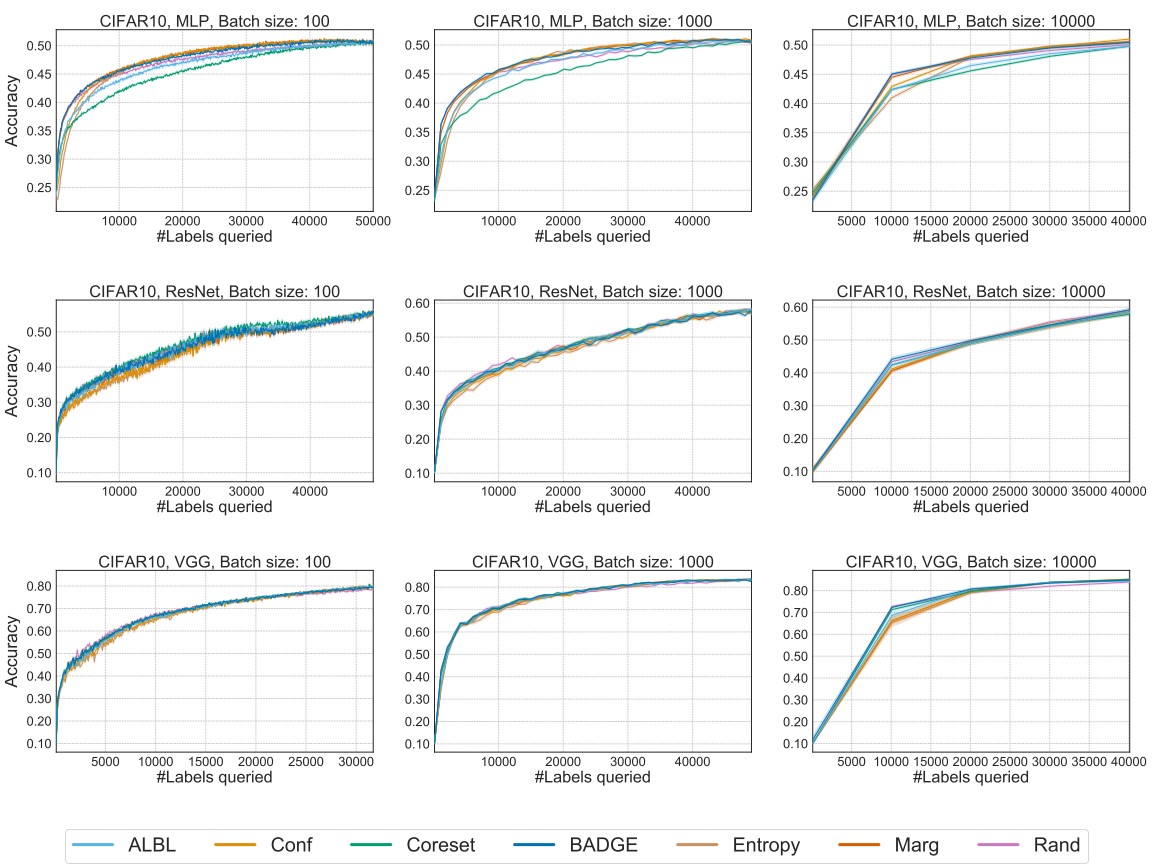

Figure 12: Full learning curves for CIFAR10 with MLP, ResNet and VGG.

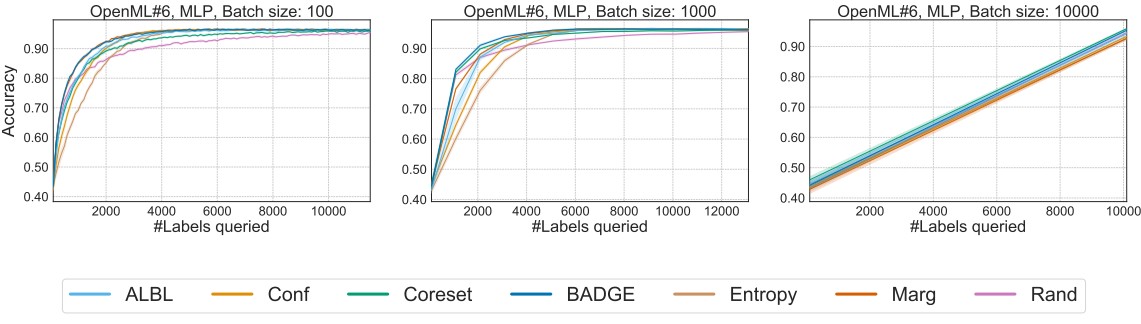

Figure 13: Zoomed-in learning curves for OpenML #6 with MLP.

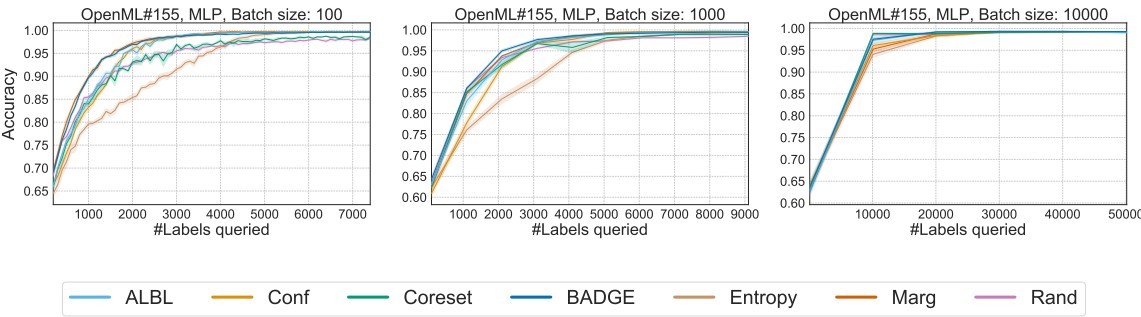

Figure 14: Zoomed-in learning curves for OpenML #155 with MLP.

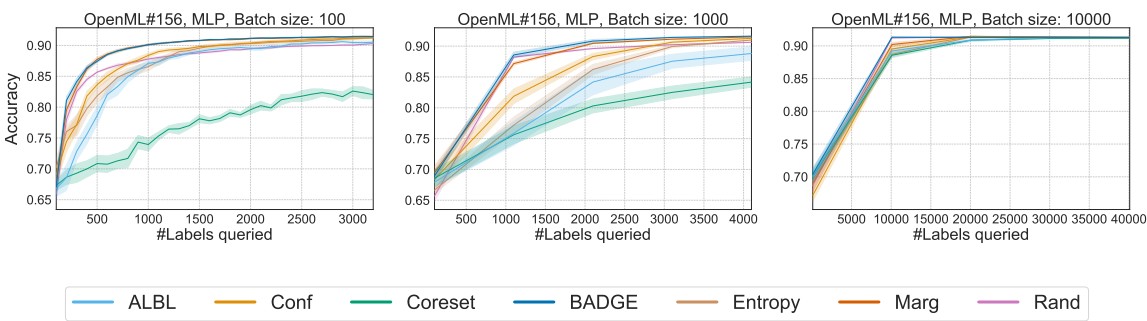

Figure 15: Zoomed-in learning curves for OpenML #156 with MLP.

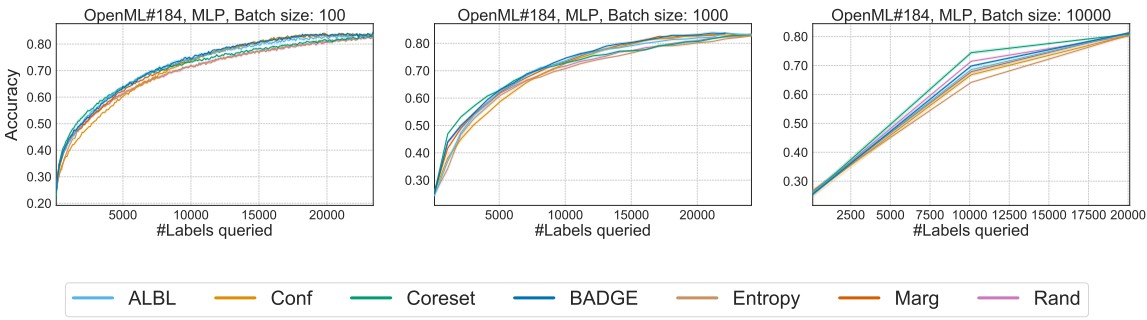

Figure 16: Zoomed-in learning curves for OpenML #184 with MLP.

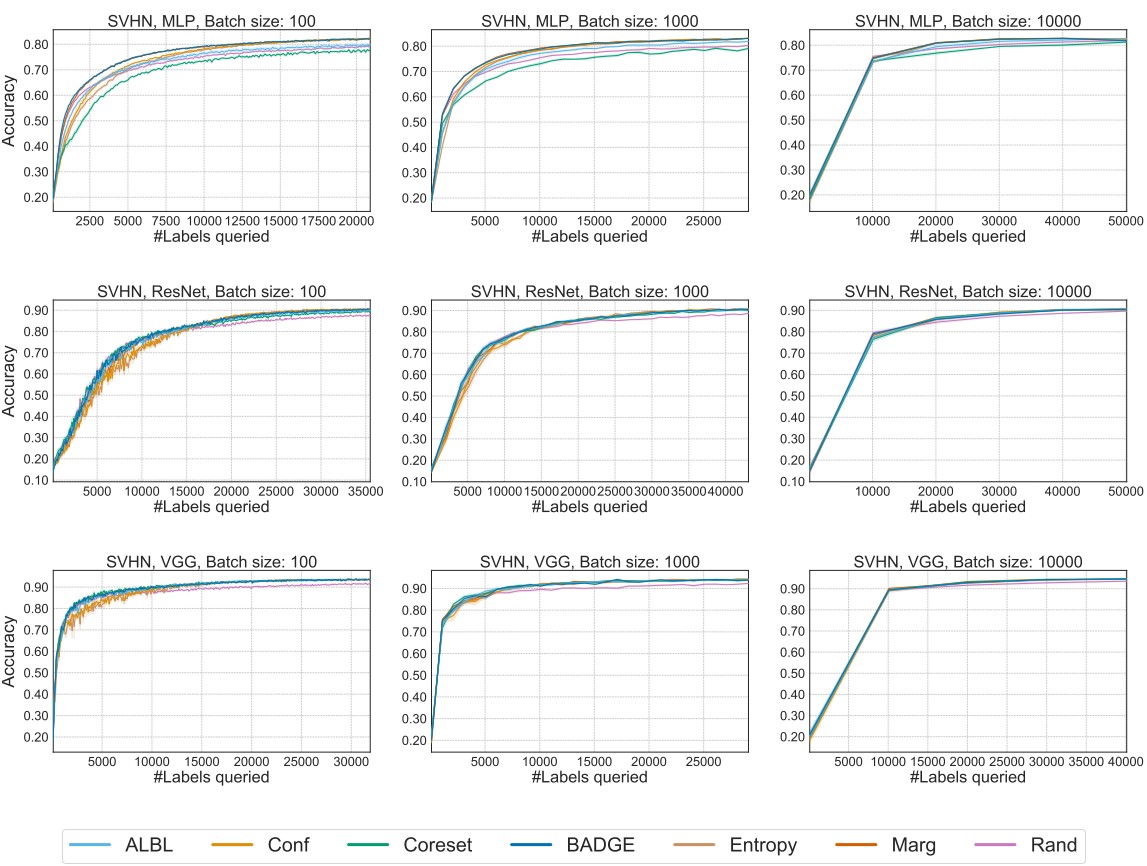

Figure 17: Zoomed-in learning curves for SVHN with MLP, ResNet and VGG.

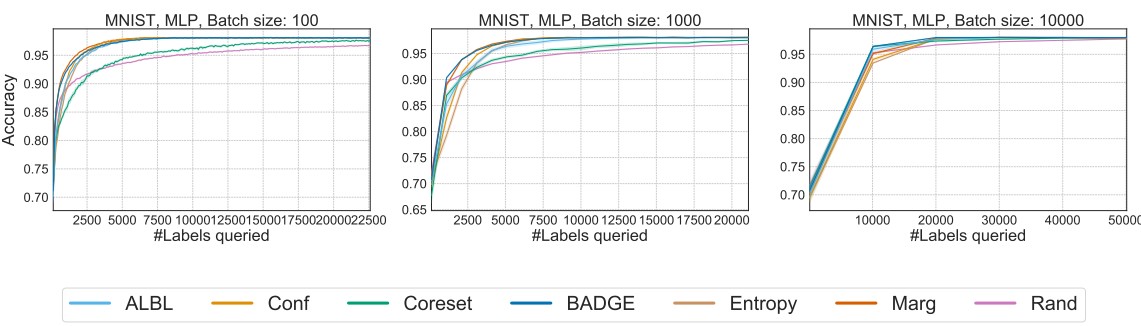

Figure 18: Zoomed-in learning curves for MNIST with MLP.

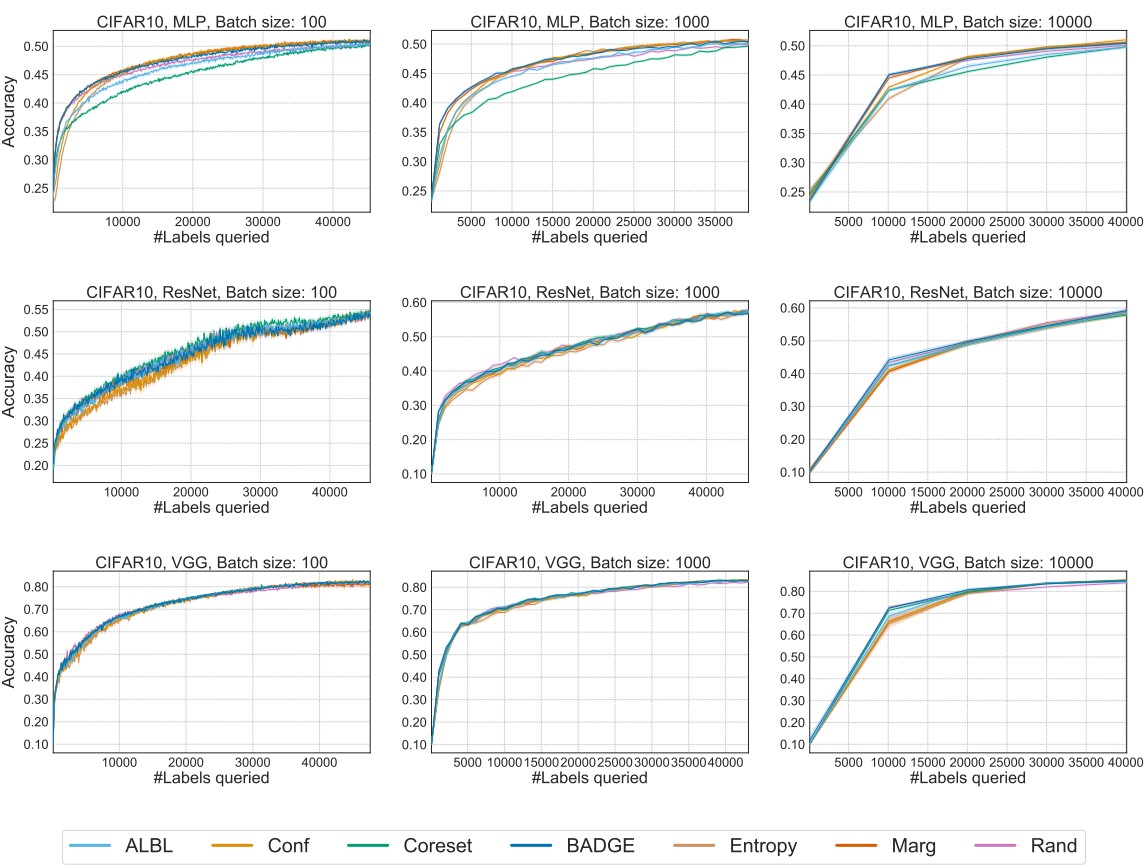

Figure 19: Zoomed-in learning curves for CIFAR10 with MLP, ResNet and VGG.

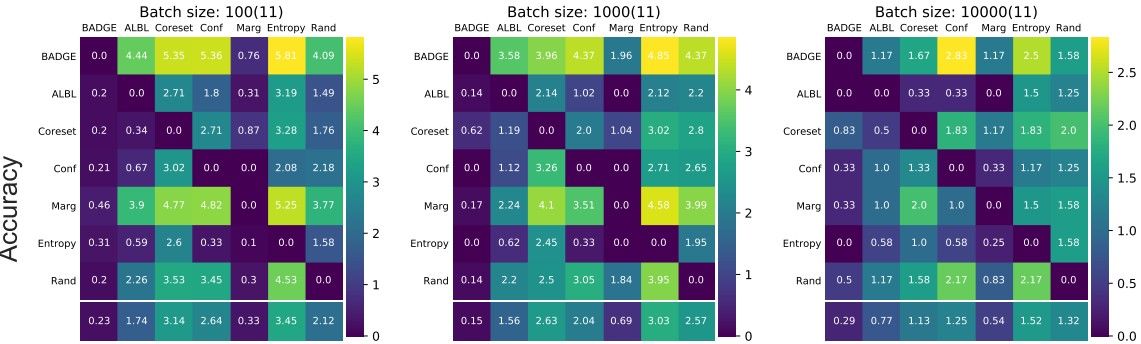

Figure 20: Pairwise penalty matrices of the algorithms, grouped by different batch sizes. The parenthesized number in the title is the total number of $(D, B, A)$ combinations aggregated, which is also an upper bound on all its entries. Element $(i, j)$ corresponds roughly to the number of times algorithm $i$ beats algorithm $j$. Column-wise averages at the bottom show aggregate performance (lower is better). From left to right: batch size = 100, 1000, 10000.

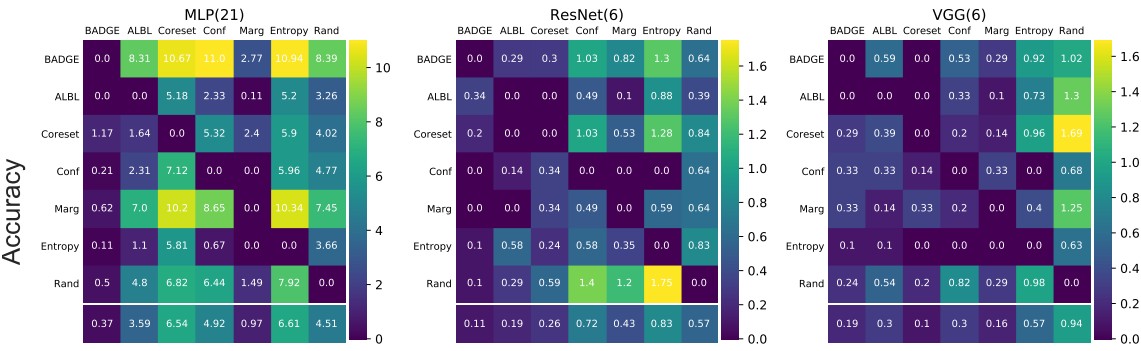

Figure 21: Pairwise penalty matrices of the algorithms, grouped by different neural network models. The parenthesized number in the title is the total number of $(D, B, A)$ combinations aggregated, which is also an upper bound on all its entries. Element $(i, j)$ corresponds roughly to the number of times algorithm $i$ beats algorithm $j$. Column-wise averages at the bottom show aggregate performance (lower is better). From left to right: MLP, ResNet and VGG.

total number of $(D, B, A)$ combinations aggregated; as discussed in Section 4, this is also an upper bound on all its entries. It can be seen that uncertainty-based methods (e.g. MARG) perform well only in small batch size regimes (100) or when using MLP models; representative sampling based methods (e.g. CORESET) only perform well in large batch size regimes (10000) or when using ResNet or VGG models. In contrast, BADGE's performance is competitive across all batch sizes and neural network models.

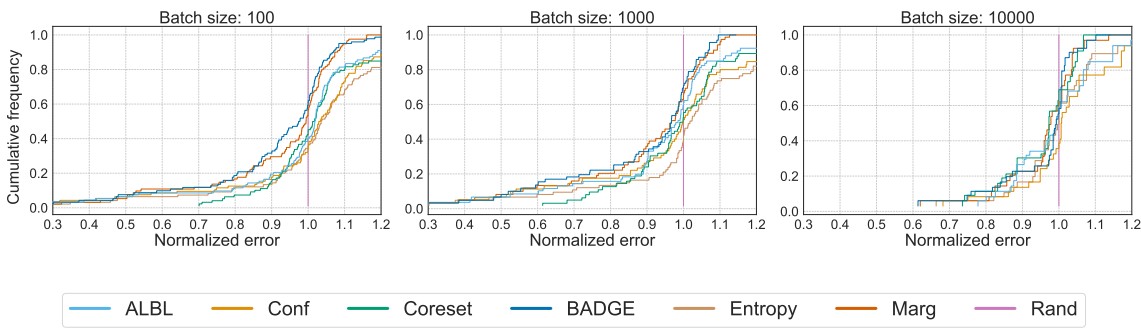

Figure 22: CDFs of normalized errors of the algorithms, group by different batch sizes. Higher CDF indicates better performance. From left to right: batch size = 100, 1000, 10000.

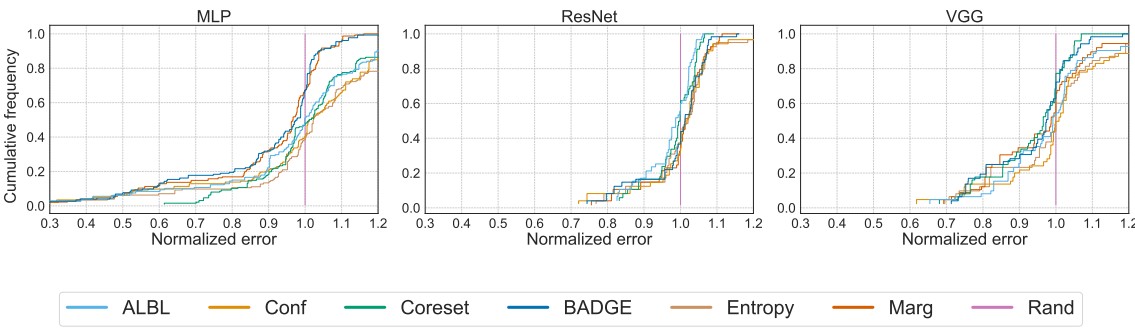

Figure 23: CDFs of normalized errors of the algorithms, group by different neural network models. Higher CDF indicates better performance. From left to right: MLP, ResNet and VGG.

## E    CDFs OF NORMALIZED ERRORS OF DIFFERENT ALGORITHMS

In addition to Figure 5 that aggregates over all settings, we show here the CDFs of normalized errors by conditioning on fixed batch sizes (100, 1000 and 10000) in Figure 22, and show the CDFs of normalized errors by conditioning on fixed neural network models (MLP, ResNet and VGG) in Figure 23.

## F    BATCH UNCERTAINTY AND DIVERSITY

Figure 24 gives a comparison of sampling methods with gradient embedding in two settings (OpenML # 6, MLP, batchsize 100 and SVHN, ResNet, batchsize 1000), in terms of uncertainty and diversity of examples selected within batches. These two properties are measured by average $\ell_2$ norm and determinant of the Gram matrix of gradient embedding, respectively. It can be seen that, $k$-MEANS++ (BADGE) induces good batch diversity in both settings. CONF generally selects examples with high uncertainty, but in some iterations of OpenML #6, the batch diversity is relatively low, as evidenced by the corresponding log Gram determinant being $-\infty$. These areas are indicated by gaps in the learning curve for CONF. Situations where there are

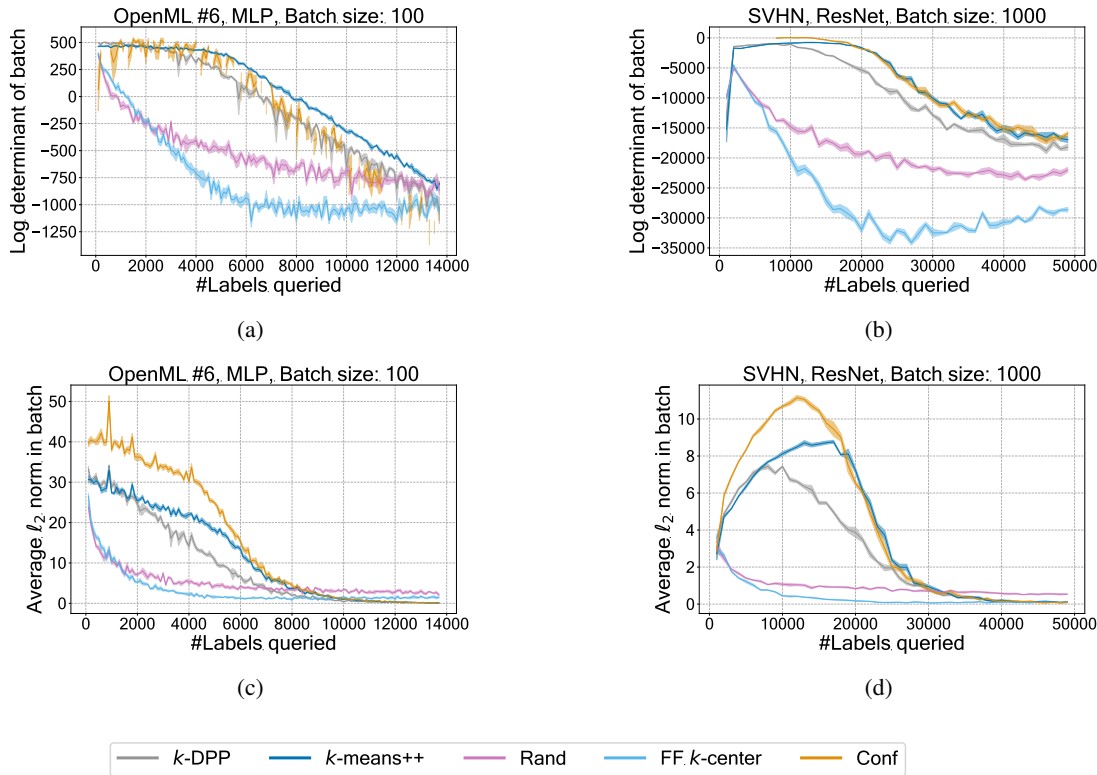

Figure 24: A comparison of batch selection algorithms in gradient space. Plots **a and b** show the log determinants of the Gram matrices of gradient embeddings within batches as learning progresses. Plots **c and d** show the average embedding magnitude (a measurement of predictive uncertainty) in the selected batch. The $k$-centers sampler finds points that are not as diverse or high-magnitude as other samplers. Notice also that $k$-MEANS++ tends to actually select samples that are both more diverse and higher-magnitude than a $k$-DPP, a potential pathology of the $k$-DPP's degree of stochastisity. Among all algorithms, CONF has the largest average norm of gradient embeddings within a batch; however, in OpenML #6, and the first few interations of SVHN, some batches have a log Gram determinant of $-\infty$ (shown as gaps in the curve), which shows that CONF sometimes selects batches that are inferior in diversity.

many gaps in the CONF plot seem to correspond to situations in which CONF performs poorly in terms of accuracy (see Figure 13 for the corresponding learning curve). Both $k$-DPP and FF-$k$-CENTER (an algorithm that approximately minimizes $k$-center objective) select batches that have lower diversity than $k$-MEANS++ (BADGE).

# G    COMPARISON OF $k$-MEANS++ AND $k$-DPP IN BATCH SELECTION

In Figures 25 to 31, we give running time and test accuracy comparisons between $k$-MEANS++ and $k$-DPP for selecting examples based on gradient embedding in batch mode active learning. We implement the $k$-DPP sampling using the MCMC algorithm from (Kang, 2013), which has a time complexity of $O(\tau \cdot (k^2 + kd))$

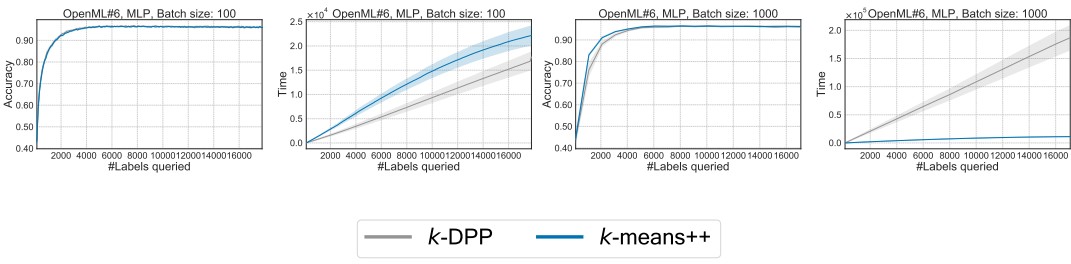

Figure 25: Learning curves and running times for OpenML #6 with MLP.

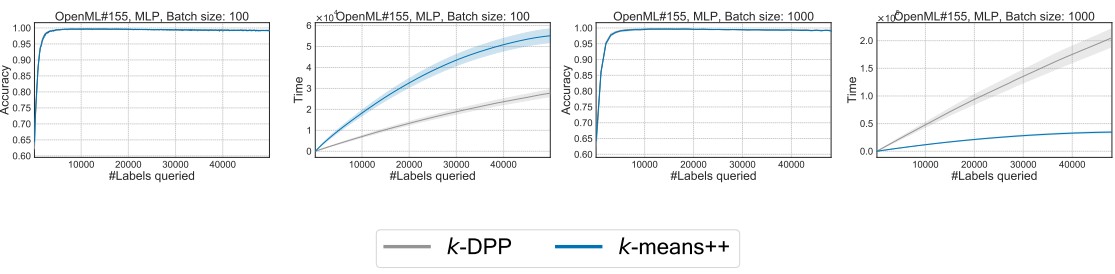

Figure 26: Learning curves and running times for OpenML #155 with MLP.

and space complexity of $O(k^2 + kd)$, where $\tau$ is the number of sampling steps. We set $\tau$ as $\lfloor 5k \ln k \rfloor$ in our experiment. The comparisons for batch size 10000 are not shown here as the implementation of $k$-DPP sampling runs out of memory.

It can be seen from the figures that, although $k$-DPP and $k$-MEANS++ are based on different sampling criteria, the classification accuracies of their induced active learning algorithm are similar. In addition, when large batch sizes are required (e.g. $k = 1000$), the running times of $k$-DPP sampling are generally much higher than those of $k$-MEANS++.

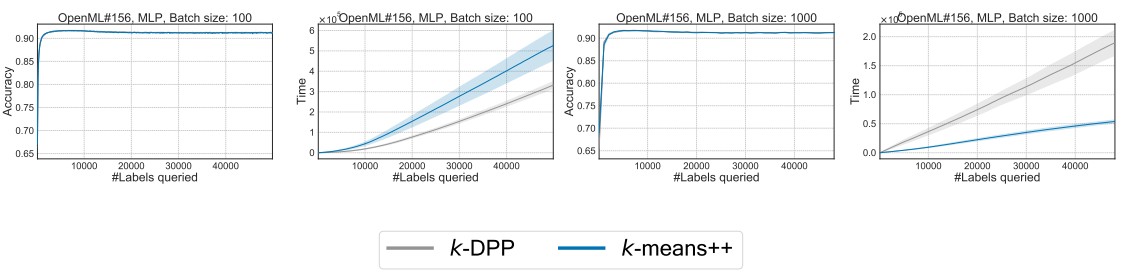

Figure 27: Learning curves and running times for OpenML #156 with MLP.

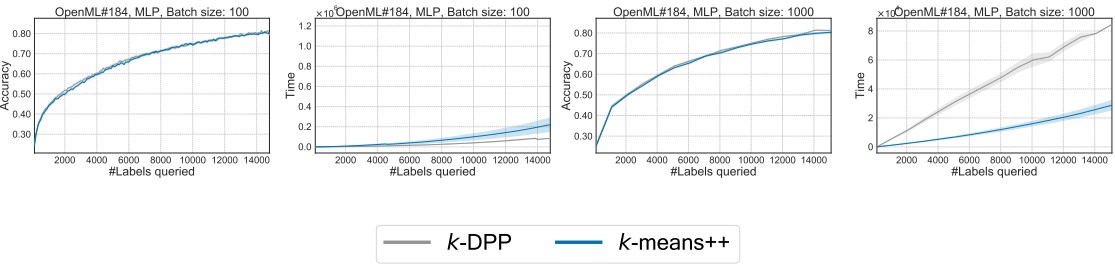

Figure 28: Learning curves and running times for OpenML #184 with MLP.

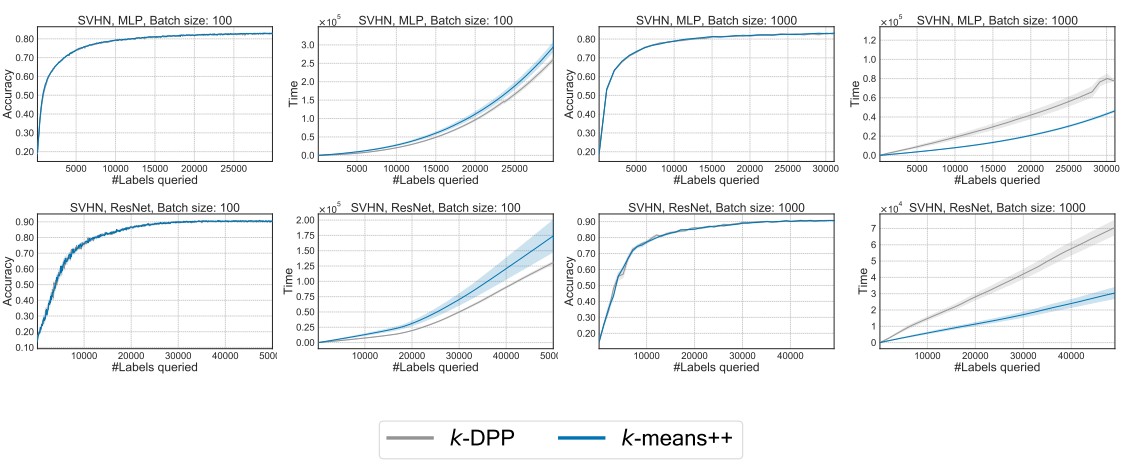

Figure 29: Learning curves and running times for SVHN with MLP and ResNet.

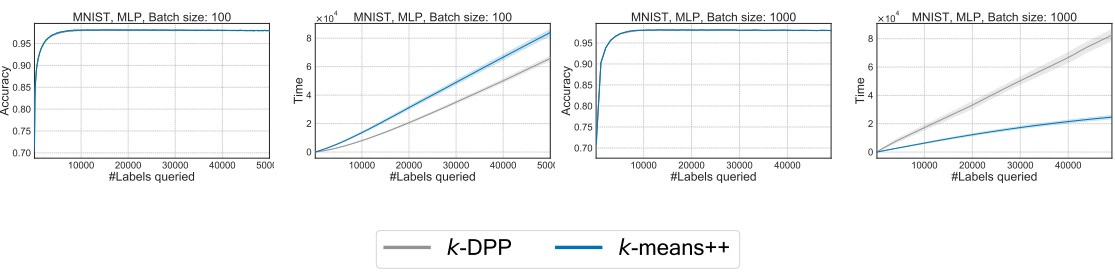

Figure 30: Learning curves and running times for MNIST with MLP.

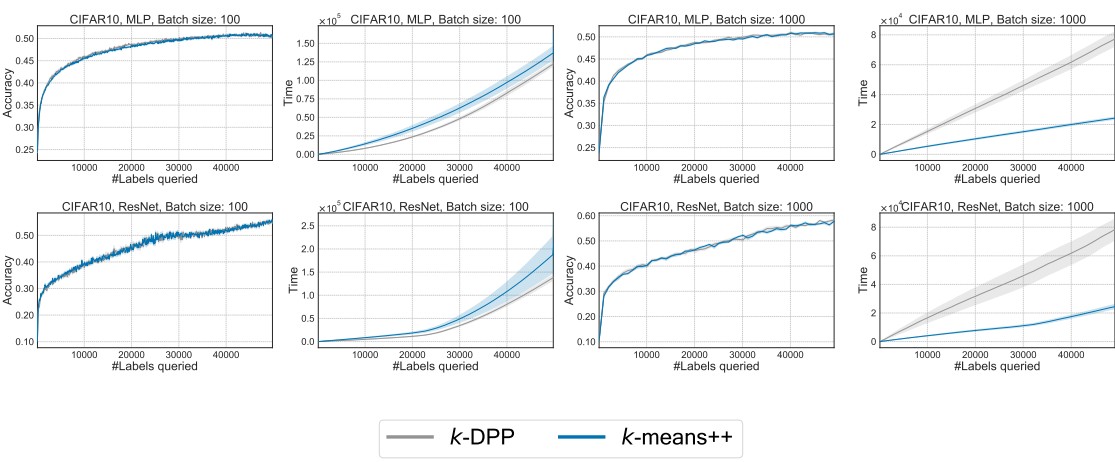

Figure 31: Learning curves and running times for CIFAR10 with MLP and ResNet.

