# OpenReview forum: "Deep Batch Active Learning by Diverse, Uncertain Gradient Lower Bounds"
_ICLR.cc/2020/Conference — Accept (Talk)_

### Official Review · AnonReviewer3 · 2019-10-22
**Official Blind Review #3**

**Rating:** 8

**Review:**

This paper introduces an algorithm for active learning in deep neural networks named  BADGE. It consists basically of two steps: (1) computing how uncertain the model is about the examples in the dataset (by looking at the gradients of the loss with respect to the parameters of the last layer of the network), and (2) sampling the examples that would maximize the diversity through k-means++. The empirical results show that BADGE is able to get the best of two worlds (sampling to maximize diversity/to minimize uncertainty), consistently outperforming other approaches in a wide-rage of classification tasks.

This is a very well-written paper that seems to make a meaningful contribution to the field with a very good justification for the proposed method and with convincing empirical results. Active learning is not my main area of expertise so I can’t judge how novel the proposed idea is, but from an outsider’s perspective, this is a great paper. It is clear, it does a good job explaining the problem, the different approaches people have used to tackle the problem, and how it fits in this literature. Below I have a couple of (minor) comments and questions:

1. Out of curiosity, it seems that it is standard in the literature, but isn’t the assumption that one can go over the whole dataset, U, at each iteration of the active learning algorithm, limiting? It is not that cheap to go over large datasets (e.g., ImageNet).
2. MARG seems to often outperform the other baselines but it doesn’t have a reference attached to it (bullet points on page 5). Is this a case that a “trivial” baseline outperforms existing methods or is there a reference missing?
3. In some figures, such as Figure 2, there are shaded regions in the plots. It is not clear what they are though. Are they representing confidence intervals? Standard deviation? They are quite tight for a sample size of 5.
4. In the section “Pairwise comparisons” it reads “Algorithm i is said to beat algorithm j in this setting if z > 1.96, and similarly … z < -1.96”.  It seems to me that the number 1.96 comes from the z-score table for 95% confidence. However, if that’s the case, it seems z should be much bigger in this context. With a sample-size of 5 (if this is still the sample size, maybe I missed something here), the normal assumptions do not hold and the t-score should’ve been used here. What did I miss?

In terms of presentation,  Proposition 1 seems to be a very interesting result. I would move it to the main paper instead of leaving it in the Appendix. I also think the paper would read better if it didn’t use references as nouns (e.g., “algorithm of (Derezinski, 2018)”). Finally, there’s also a typo on page 7 (Apppendx).


---

>>> Update after rebuttal: I stand by my score after the rebuttal.  This is a really strong paper in my opinion. I appreciate the fact that the authors took my feedback into consideration.

**Experience Assessment:**

I do not know much about this area.

**Review Assessment: Checking Correctness Of Derivations And Theory:**

I assessed the sensibility of the derivations and theory.

**Review Assessment: Checking Correctness Of Experiments:**

I assessed the sensibility of the experiments.

**Review Assessment: Thoroughness In Paper Reading:**

I read the paper thoroughly.

---

> ### Author Response · Authors · 2019-11-15
> **Response to Reviewer 3**
>
> Thanks for your feedback.
>
> 1. Yes, it is standard in the literature to use the entire unlabeled dataset.  To deal with large pool sizes, one can perform subsampling and use our approach to select examples for label queries within the subsample.
>
> 2. The earliest citation we can find for MARG is from “Margin-based Active Learning for Structured Output Spaces” by Roth and Small. We’ve added the citation.
>
> 3. Shaded regions are standard error. We added that to figure captions.
>
> 4. Yes, we agree that a t-test would be more appropriate here (although both are not ideal, as the distributions of the error differences can be far from Gaussian). We have updated our comparison matrices in light of this discussion.

---

### Official Review · AnonReviewer1 · 2019-10-22
**Official Blind Review #1**

**Rating:** 6

**Review:**

The paper proposes a new method for active learning, which picks the samples to be labeled by sampling the elements of the dataset with highest gradient norm, under some constraint of diversity. The aforementioned gradient is computed w.r.t. the predicted label (rather than the true label, that is unknown) and diversity is achieved by sampling via the k-MEANS++ algorithm.
The paper is well written and while the experiments look thorough, the motivation to support the proposed method seem too weak and unconvincing as does the discussion of the results, which is why I am leaning toward rejection.
I am willing to amend my vote if the authors provide stronger (not empirical) motivations on why using the gradient norm w.r.t. the predicted label is a better metric than those in the literature, and  More comments below.

Detailed feedback:

1) The paper lacks a proper motivation as to why using the norm of the gradient is a better metric than the many others already present in the literature. In particular, I cannot think of any case where it would be best to use that than the entropy of the network’s output distribution, even though the empirical results seem to suggest otherwise. Specifically, while I believe that in many cases it will be similarly good, if we consider the case when the network is able to rule out most of the classes but is unsure on a small fraction of them, the entropy will better reflect this uncertainty than the norm of the gradient of the predicted class.

Generally speaking, I believe that the use of the norm of the gradient of the predicted class should be much better motivated, being the core idea of the paper. Stating that it is cheap to compute and empirically performs as well as k-DPP in two experiments is not convincing enough in my opinion.
2) I wonder how much of the performance of BADGE is due to k-MEANS++ and how much to the choice of using the gradient norm. Please perform an ablation study where you can e.g., replace the gradient norm with the entropy, or replace k-MEANS++ with random sampling, and discuss the results.
3) How is the embedding “ground set” space determined for k-MEANS++? How are the centroids determined? In which space? It is unclear to me how k-MEANS++ is used in the context of the norm of the gradients. Please improve the explanation in the main text.
4) Please add a curve for k-DPP to the plots in the main text, rather than having separate plots for it in the appendix. Also, it would be interesting to compare against Derezinski, 2018 as well, if that’s the current state of the art (which is what I infer from your text, but I might be wrong).
5) The paper builds on the claim that the gradient norm w.r.t. the prediction is a lower bound for the gradient norm induced by any other label, yet Proposition 1 that proves it is in Appendix B. This prove is central to the proposed idea and should be in the main text.
6) The authors claim that to capture diversity they collect a batch of examples where the gradients span a diverse set of directions, but it’s unclear to me that k-means++ actually accomplishes that. Where is the *direction* of the gradient taken into account in the algorithm?
7) The “discussion” section is really a “conclusion” one, and indeed a proper in-depth discussion of the experiments is missing. Please expand the comments on the experimental results.
8) The metric to compute the “pairwise comparison” looks quite convoluted. Is it common in the literature? If so, please add a reference. If not, can you motivate the use of this specific formula?
9) The random baseline seems to be very competitive. Why is that? Please provide your intuition. Could this be indicative that the baselines have not been tuned properly?
10) Introduction: the sentence “[deep neural networks] successes have been limited to domains where large amounts of labeled data are available” is incorrect. Indeed, neural networks have been used successfully in many domains where labelled data is scarce, such as the medical images domain for example. Please remove the sentence.
11) Introduction: please add a sentence to explain what a version-space-based approach is.

12) Is Figure 2 the average over multiple runs or a single run?

13) Notation: please do not use g for the gradient (g^y_x) and for the intermediate activations (g(x; V)).

14) The lower margin seem too wide. Please make sure you respect the formatting style of the conference.


Minor:
- Notation: if you must shorten g^{\hat{y}}_{x} please do so with \hat{g}_{x} and equivalently shorten g^{y}_{x} as g_{x}
- Notation: in the pairwise comparison, please don’t reuse i to denote an algorithm (it is used a few lines before to compute the labeling budget)
- Please add reference to Appendix A when k-MEANS++ is first referred to in page 2.
- Page 3,  when Proposition 1 is mentioned add reference to the location where it’s defined.


Typos:
- Page 2: expenive -> expensive
- Page 5: Learning curves. “Here we show ..” -> Remove “here”
- Figure 3: pariwise -> pairwise
- Page 7: Apppendx E


----------------------
Updated review:

I thank the authors for for taking the time to address all my comments, and clarifying some of the misunderstandings I had. I am happy to revise my score accordingly.

**Experience Assessment:**

I do not know much about this area.

**Review Assessment: Checking Correctness Of Derivations And Theory:**

I carefully checked the derivations and theory.

**Review Assessment: Checking Correctness Of Experiments:**

I carefully checked the experiments.

**Review Assessment: Thoroughness In Paper Reading:**

I read the paper thoroughly.

---

> ### Author Response · Authors · 2019-11-15
> **Response to Reviewer 1**
>
> Thanks for your effort.
>
> 1. We believe you may have missed the main contribution of this paper - the nature of the embedding used. We do not use the hallucinated gradient norm as the embedding, we use the gradient vector for the parameters at the last layer of the network. Once samples are embedded in this space, which is of dimension equal to (number of classes x number of penultimate layer nodes), we use the k-means++ algorithm to select a batch of k representative samples. We note that the norm of this vector is a lower-bound on the true norm of the last-layer gradient, given the corresponding label to a given sample.
>
> 2. Again, we are not using the gradient norm as our embedding. The gradient representation is not interchangeable with an uncertainty metric like entropy.
>
> We do use random sampling as a baseline. Random sampling is not conditioned on any representation.
>
> 3, 6. The centroids are the output of the k-means++ algorithm. Again, this is run in the space of the gradient embeddings, not the space of gradient norms. Each embedding has both direction and magnitude.
>
> 4. We do have this in the main text. Representative k-DPP plots are shown in Figure 1 in comparison to k-means++. The reason we do not present results for k-DPP in Section 4 (Experiments) is that sampling from k-DPP is much more time consuming than all other methods considered, while its performance is similar to kmeans++. This is why we use kmeans++ in BADGE.
>
> 5. We added this proof to the main text.
>
> 7. Our explanations of Figures 1-5 describe experimental trends. We describe experimental results in terms of the behavior of plots, especially in the Experiments section. Some explanation was added in the most recent article update.
>
> 8. Unfortunately, besides examining learning curves like those in Figure 2, there are no widely-used metrics for evaluating batch active learning in the literature. We choose this metric because we are interested in which algorithms significantly outperform other algorithms for various labeling budgets. In the current version of the paper, the comparison comes from a t-test.
>
> 9. Diversity-based approaches often perform worse-than-random when the penultimate layer representation is not meaningful. That is, because random sampling is not conditioned on any representation, it can actually induce a more diverse batch. We also sometimes see random outperforming confidence-based approaches, which is evidence that selecting on diversity is better than selecting on uncertainty for those situations.
>
> None of the baseline acquisition functions have tunable parameters.
>
> 10. We’ve weakened the claim in the first sentence.
>
> 11. The version space is the space of all models that are plausible given the labeled example seen so far. We’ve included that in the introduction.
>
> 12. Each line in figure 2 is averaged over five runs. The shadow for each line describes the standard error over those runs. We’ve added this to the text.
>
> 13. We changed the intermediate activation function to z(x; V) to avoid confusion.

---

### Official Review · AnonReviewer2 · 2019-10-23
**Official Blind Review #2**

**Rating:** 8

**Review:**

Batch active:
This paper proposes a novel approach to active learning in batches. Assuming a neural-network architecture, they compute the gradients of each unlabeled example using the last layer of the network (and assuming the label given by the network) and then choose an appropriately diverse subset of these using the initialization step of kmeans++. The authors provide intuitive motivation for this procedure, along with extensive empirical comparisons.

Overall I thought the paper was well written and proposed a new practical method for active learning. There were a few concerns and places where the paper could be clearer.

1. The authors keep emphasizing a connection to k-dpp for the sampling procedure emphasizing diversity. They provide a compelling argument for the kmeans++ but in Figure 1 it is unclear why k-DPP is the right comparison point. For example, you could imagine building a set cover of the data using balls at various radii and then choosing their centers.
2. The paper emphasizes choosing samples in a way to eliminate pathological batches. Considering this is a main motivation, none of the figures really demonstrate that this is what BADGE is doing compared to the uncertainty sampling-based methods tested against. Perhaps the determinant of the gram matrix of the batch could be reported for both algorithms?
3. While reading the paper, the set of architectures used was hard to find. Maybe I just missed it, but it would be useful to have this information. In particular, in Figure 3, there are absolute counts, but I wasn’t sure how many (D,B,A,L) combinations there were.
4. Finally, recent work in Computer Vision has shown that uncertainty sampling with ensemble-based methods in active learning tends to work well. I understand that it is hard to compare to the myriads of active learning algorithms out there, but they deserve a mention. See [1] below.

Overall I think this paper is a good empirical effort that I recommend for acceptance.

[1] Beluch, William H., Tim Genewein, Andreas Nürnberger, and Jan M. Köhler. "The power of ensembles for active learning in image classification." In Proceedings of the IEEE Conference on Computer Vision and Pattern Recognition, pp. 9368-9377. 2018.

**Experience Assessment:**

I have published one or two papers in this area.

**Review Assessment: Checking Correctness Of Derivations And Theory:**

I assessed the sensibility of the derivations and theory.

**Review Assessment: Checking Correctness Of Experiments:**

I assessed the sensibility of the experiments.

**Review Assessment: Thoroughness In Paper Reading:**

I read the paper at least twice and used my best judgement in assessing the paper.

---

> ### Author Response · Authors · 2019-11-15
> **Response to Reviewer 2**
>
> Thanks for your review.
>
> 1-2. The motivation for using a k-DPP is that it will select a batch of samples that are both high magnitude and diverse. K-means++ has this property too - in particular, if initialized with a high-magnitude point, proceeding samples are likely to be high-magnitude as well. We show this phenomenon in Figure 2 of the newly-updated paper, and compare it to another method for clustering data. We also added appendix figures to appendix F, showing that simple uncertainty sampling can lead to batches with Gram determinant zero.
>
> 3. We used three architectures (ResNet, VGG, and MLP), seven datasets (MNIST, SVHN, CIFAR-10, and four OpenML datasets), and three different batch sizes (100, 1k, and 10k). We didn’t use any convolutional architectures with MNIST or non-image datasets, leading to 33 unique combinations of dataset, batch size, and architecture. As each (dataset, batch size, architecture) combination only contribute to at least a penalty of 1 in the penalty matrix, the largest entry in the penalty matrix is at most 33. We made that clear in the newly-updated copy.
>
> 4. Thank you for pointing out this article. We’ve added it to the related work section.

---

### Author Response · Authors · 2019-11-15
**To All Reviewers**

Thank you all for your review. We've changed some notation, fixed typos, and moved Proposition 1 to the main text as per your recommendations. We respond to your individual comments below.

---

### Decision · Program_Chairs · 2019-12-19

**Decision:**

Accept (Talk)

**Comment:**

The paper provides a simple method of active learning for classification using deep nets. The method is motivated by choosing examples based on an embedding computed that represents the last layer gradients, which is shown to have a connection to a lower bound of model change if labeled. The algorithm is simple and easy to implement. The method is justified by convincing experiments.

The reviewers agree that the rebuttal and revisions cleared up any misunderstandings.

This is a solid empirical work on an active learning technique that seems to have a lot of promise. Accept.